# The Design Value for Recycling End-of-Life Photovoltaic Panels

Michele Calì [1,*], Bekkay Hajji [2], Gioele Nitto [1] and Alberto Acri [3]

1   Electric, Electronics and Computer Engineering Department, University of Catania, 95125 Catania, Italy
2   Laboratory of Renewable Energy, Embedded System and Information Processing, National School of Applied Sciences, Mohammed First University, Oujda 60000, Morocco
3   Department of Engineering, University of Messina, 98158 Messina, Italy
*   Correspondence: michele.cali@unict.it; Tel.: +39-0957382400

**Abstract:** The production of electric energy has been increasingly deriving from renewable sources, and it is projected that this trend will continue over the next years. Among these sources, the use of solar energy is supposed to be considered the main future solution to global climate change and fossil fuel emissions. Since current photovoltaic (PV) panels are estimated to have an average life of 25–30 years, their disposal is very important for the recovery of materials already used and for introducing them again into other processing cycles. Innovative solutions are therefore needed to minimize the emissions of pollutants derived from the recycling of photovoltaic panels that no longer work. In this research, an analysis of data related to durability, recyclability rates, different possible design layouts and materials used in the design and manufacture of PV panels was conducted. Through a Design for Recycling (DfR) and a Design for Durability (DfD), the authors identified the optimal materials, the best geometries and geometric proportions as well as the most convenient geometric and dimensional tolerances in the couplings between the layers and the components that comprise the panel to attain the most current, efficient and effective solutions for recycling end-of-life (EoL) PV panels and for longer durability.

**Keywords:** recyclability rates; end-of-life management; EcoDesign method; coupling tolerances; sustainability

## 1. Introduction

The use of renewable and sustainable energy has widely been proposed to reduce environmental degradation to try to limit climate change significantly. In particular, we are detecting a new interest in the possibility of generating electricity through PV panels. In order not to cause environmental impacts especially inherent to the use of agricultural land, we derived the idea to apply PV panels in industrial basins, irrigation tanks and drinking water tanks using floating PV (FPV) [1,2]. These FPV installations represent new opportunities for the diffusion of PV, especially in countries with a high population density and with little land available.

In 2019, the European Commission published the European Green Deal [3], a document supplied to face climate changes through a precise series of targeted actions. To reach the goal, Europe must no longer generate net greenhouse gas emissions by 2050. Moreover, the development model must change itself to dissociate economic growth from the waste of resources. It is an ambitious project, which will affect tens of millions of people and on which all the main European institutions will be hard at work for years. If the pact is well implemented, Europe will be the first continent to have achieved climate neutrality [4]. Today, however, the Green Deal seems to be a big question because of the Russia–Ukraine conflict. Some plans exist to start by using coal for energy generation due to the lack of natural gas from Russia, and there is no important plan with a "renewable" solution in place in the coming years.

Figure 1, taken from the Commission Document, illustrates the main elements of the European Green Deal [5]. The document of the European Green Deal substantially aims to promote the efficient use of resources to have a clean and circular economy, stop climate change, restore biodiversity loss and reduce pollution [6].

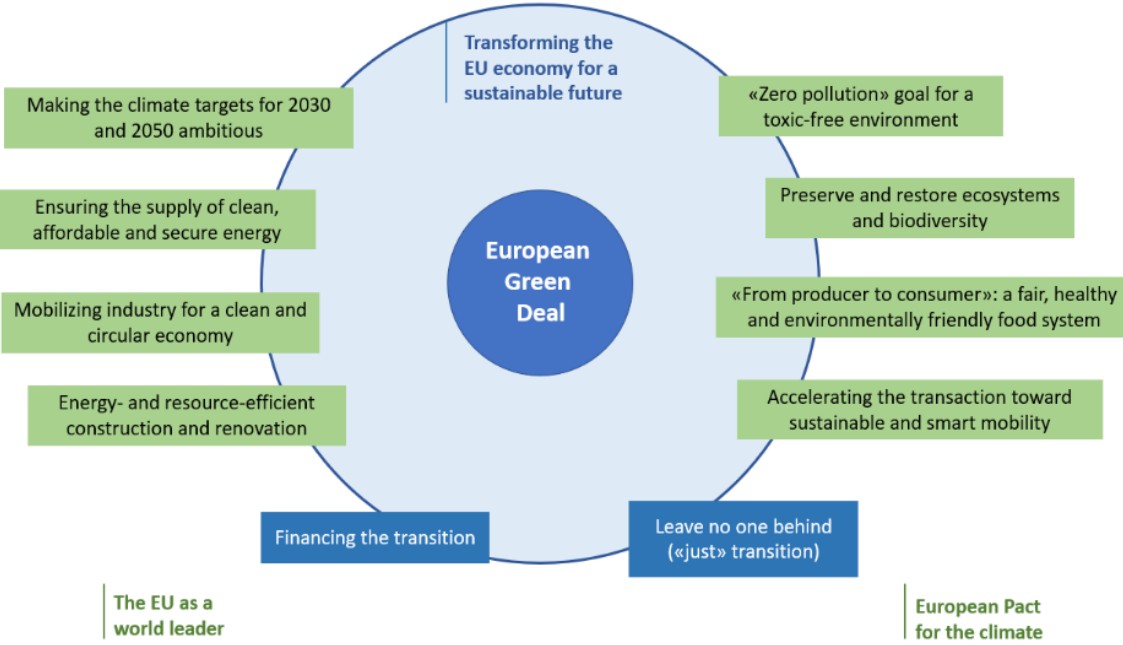

**Figure 1.** Various elements that comprise the European Green Deal.

The achievement of these objectives can be obtained specifically by creating recyclable and biodegradable materials that not only have a minimal impact in environmental terms but that are also designed to be reused several times to avoid losing their original characteristics [4,5].

It follows that, to generate a sustainable economy, it seems to be necessary to consider the link between EcoDesign and Circular Economy.

**EcoDesign** (or sustainable design/planning) is an economic model that involves the entire process of conceiving, designing, selling on the market and the disposal of a product that respects the environment, through reductions to the minimum levels of the negative impact that it could have on the ecosystem. Consequently, the materials chosen must be sustainable and recyclable with the utmost respect for the environment and primary resources.

**Circular Economy** is a model of production and consumption that involves sharing, lending, reusing, repairing, reconditioning and recycling existing materials and products for as long as possible [7–9].

In light of these considerations, we can say that EcoDesign is the first step towards a Circular Economy, as it considers the environmental impact that a given product has throughout its life cycle, from its conception to its disposal. In fact, EcoDesign and Circular Economy together constitute the foundations of a sustainable economy.

Europe has important infrastructure for the production of solar and wind energies and for the storage of energy and the use of portable batteries. Since the infrastructure is being replaced by more modern structures and the maintenance cycle requires a possible replacement of parts, the application of Circular Economy principles is the basis for exploiting the resource potential of the waste generated and for minimizing the challenges of their management. Some studies have estimated that hundreds of thousands of tons of old wind turbines, batteries and solar panels must be disposed of or recycled in the next decade along with millions of tons by 2050 (Figure 2) [10].

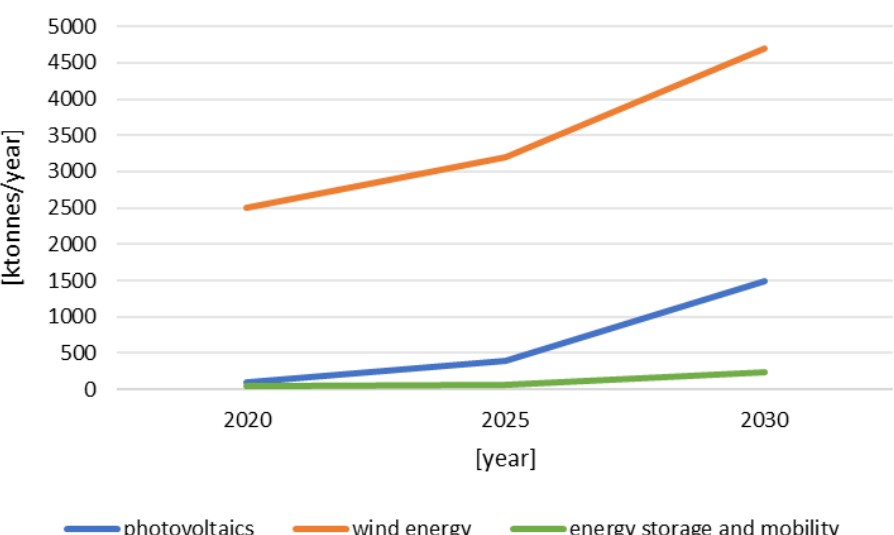

**Figure 2.** Growth of waste materials from renewable resources.

The biggest challenge is recovering materials from decommissioned devices. Therefore, we should try to design and manufacture devices that do not break and that can carry out their function efficiently for as long as possible, which is a central aspect. To date, PV panels have a lifespan of 25–30 years. In recent years, we have been working to efficiently recover the materials of solar panels at the end of their life cycle to ensure convenient investment in recycling plants. The producers, in fact, think in terms of profitability and not recyclability. It is, therefore, necessary to make recycling economically advantageous or, at least, to encourage it with government contributions. To date, silicon-based panels are the most common but only for reasons of producer profitability.

In Europe, the process to achieve complete recycling of EoL products has already started. With Directive 2008/98/EC on waste, the EU has entrusted the producer itself with responsibility for its EoL panels, allowing the producer to include in the price initial costs for the EoL treatment. In 2012, Directive 2012/19/EU introduced the first regulation on disposal and recycling, opening the doors to different financing models for the separate collection of PV panels.

In this research, the authors analyze data on the durability, recyclability rates, different possible design layouts and materials used in the design and manufacture of PV panels. The analyzed data were provided by the *Laboratory of Renewable Energy, Embedded System and Information Processing*, National School of Applied Sciences, Mohammed First University, Oujda 60000, Morocco. Through analytical evaluations and numerical simulations (Dfr and DfD approach), the study identifies the optimal materials, the best geometries and geometric proportions and the most convenient geometric and dimensional tolerances in the couplings between the layers and the components that comprise the panels with the aim of achieving the most current, efficient and effective solutions for recycling EoL PV panels and for longer durability.

Finally, a logical scheme is proposed that allows the total recycling of materials at the end of their life (EoL) for panels designed and built in the optimized way.

The paper is organized as follows. Section 2 describes the opportunities and challenges for PV panel recycling. In Section 3, after a preliminary description of the structure and materials that constitute PV commercial panels, an effective design method for recycling and incrementing durability is presented, and a logical scheme that allows the total recycling of materials at the EoL is proposed. In Section 4, a numerical analysis of the proposed method is carried out, and the obtained results are discussed. Finally, in Section 5, the conclusions are shown.

Figure 3 shows all the main steps of the present research.

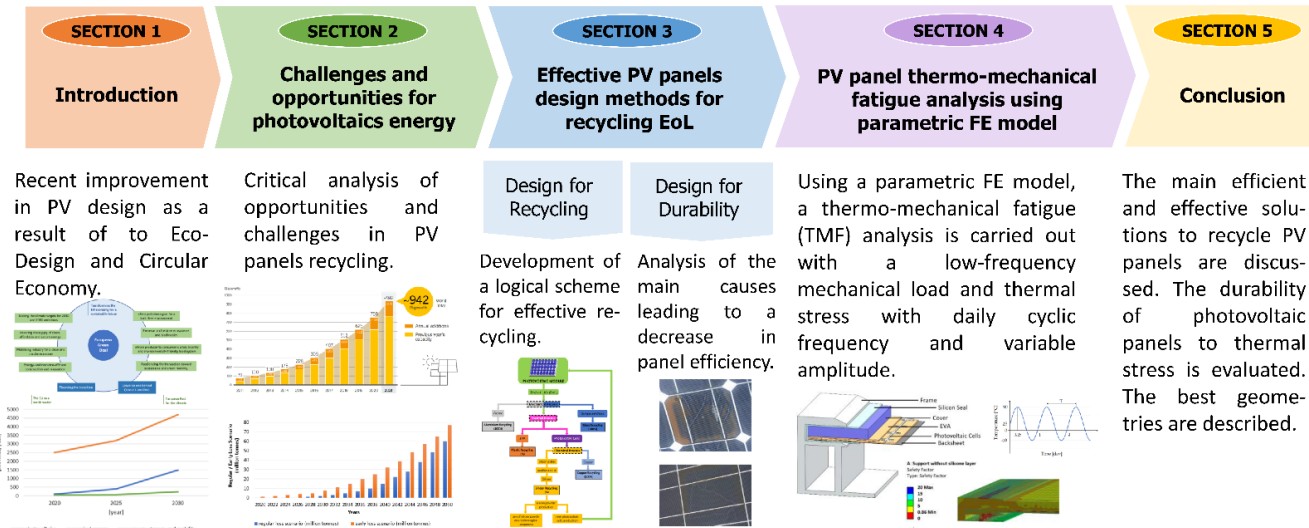

**Figure 3.** Main research steps.

## 2. Challenges and Opportunities for Photovoltaic Energy

The use of energy from PV panels is considered a "low environmental impact" choice since it exploits a 'clean' source such as the sun, avoiding the production of greenhouse gases. In 2018, photovoltaics began to be the fastest growing energy technology in the world. According to the latest authoritative reports [11–13], the market for PV panels has been continuously increasing. PV electricity capacity has continuously grown exponentially with only a few disturbances, mainly due to strong increases in the costs of raw materials and shipments. As a result of new installations, in 2021, there was a cumulative global capacity of 942 GW (Figure 4). In 2021, there was a total annual addition of 175 GW, with an increase of 36 GW compared to 2020.

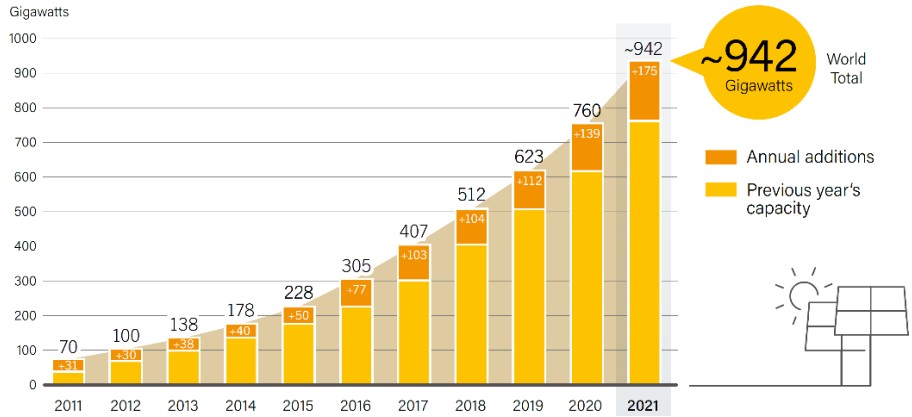

**Figure 4.** Exponential growth of PV energy in the years 2011–2021 (source [11]).

The leading countries for cumulative PV capacity are China, the United States, Japan, India and Germany, and the leading countries for per capita capacity are Australia, the Netherlands and Germany (Figure 5). Currently, of all the PV panels installed globally, China holds the largest share with 31%, followed by the United States (15%) and India (7%) [11].

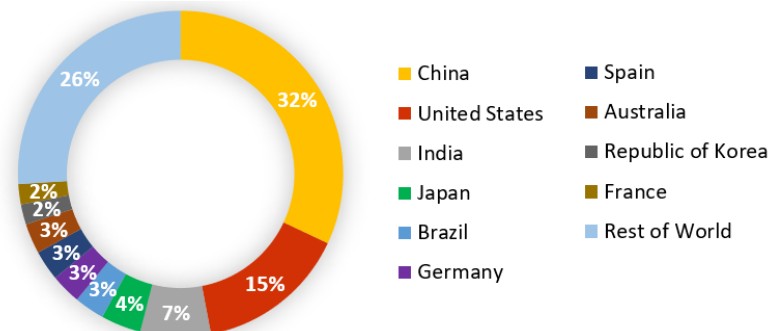

**Figure 5.** PV panels globally installed in 2021.

However, the increase in the installation of PV panels means an increase in the amount of waste produced when these panels reach their EoL. Considering the study of Chowdhury et al. [14], a dizzying increase will be registered in metric tons per year of PV panel waste over the next two decades. In fact, it is expected that the European Union will generate 500,000 tons of waste per year compared to the current 30,000 tons.

Compared to the current situation of PV panels installed (Figure 4), some authors [15] estimate an exponential increase, especially in Europe and in Italy, resulting in a dramatic increase in waste from PV between now and 2050 (Table 1).

**Table 1.** Estimate of tons of waste from PV panels in Europe and Italy.

| Year | 2013 | 2020 | 2030 | 2040 | 2050 |
|---|---|---|---|---|---|
| **Europe** | 11,395 | 33,000 | 133,000 | 4,000,000 | 9,500,000 |
| **Italy** | 1757 | 1000 | 5000 | 1,000,000 | - |

Several studies [16–19] highlight the critical issues of PV panels in EoL phases due to the risk of releasing highly polluting substances into the soil, groundwater or the atmosphere. This is particularly true if the modules are disposed of as they are in landfills or if they undergo incineration processes, in which they can release highly polluting substances into the soil, groundwater or the atmosphere. Hence, it is important to develop adequate EoL management processes through appropriate disposal/recycling technologies in an environmentally and economically sustainable way.

Today, technologies capable of recycling 95–99% of PV panels materials (e.g., glass, copper, aluminum, etc.) do exist. Most of the recyclable materials in PV panels are based on glass with about 68% by weight, aluminum with about 15% by weight, high-purity silicon with about 3% by weight and copper cables with about 1% by weight (Figure 6).

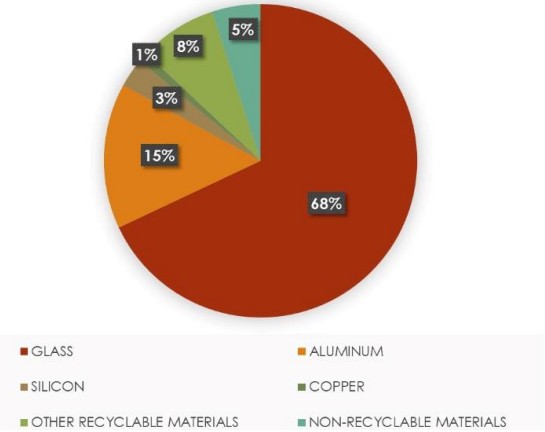

**Figure 6.** Recyclable materials in PV panels.

With reference to the percentage of recyclability of materials, different recycling methods achieve different results. It is worth remembering that some elements are present in very low percentages. The opportunity of their recovery at their EoL depends not only on the technical possibility of carrying out the action, but also on the cost/benefit of the process in economic terms and therefore on the correct design with which the panels are designed, foreseeing their entire life cycle. Finally, it is necessary to highlight the considerable benefits that are obtained by increasing the average life (durability) of the panels while maintaining high efficiency.

Figure 7 highlights the accumulation of non-recycled PV panels in the case that their useful life is 30 years. The bar chart shows two aspects: in a scenario of regular loss (PV panels operating up to 30 years), a dramatic increase of 60 million tons is expected by the end of 2050, and in an early loss scenario (PV panels failing before the age of 30), waste is expected to rise to 80 million tons by 2050 [19].

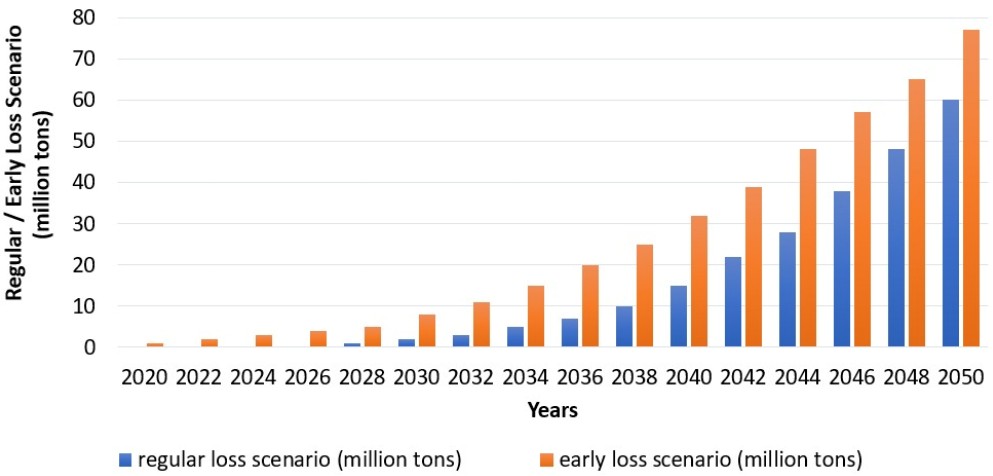

**Figure 7.** Amount of waste produced by PV panels for 25 or 30 years of average durability.

In general, a distinction can be made between low-value recycling solutions, which are aimed at the recovery and recycling of some fractions of the module's bill of materials, such as glassy and frame aluminium, and high-value recycling solutions, which make it possible to maximize the value of waste derived from end-of-life modules, recovering also the most valuable components in them. The latter are represented by silicon (contained in crystalline silicon modules), semiconductors used in thin-film panels, and silver, mainly used by crystalline silicon technologies. These materials are conceived to be enhanced and reinserted into new production processes.

The recovery of the metal components of the panels is complex and requires additional treatments, which are expensive and have an impact on the environment, and the recovery of plastics is of little value. The major manufacturers of PV panels have experimented with different methods for the separation and recovery of various materials derived from PV panels at the end of their life. The various technologies that were investigated can be divided into types: physical treatments (shredding with separation by density or magnetism of metals), chemical treatments (acid/alkaline attack or dissolution with organic solvents) and thermal treatments (pyrolysis, incineration and melting of polymeric materials).

To date, the total recycling of all components is not practiced. Numerous research projects underway (RESIELP recovery of silicon and other materials from EoL PV panels, PHOTORAMA, etc.) are moving in this direction.

The most recent technologies make it possible to extract 99% of the high-value metals contained in photovoltaic solar panels (silver, silicon, copper and aluminum) and to reuse or return them to the supply chain [19]. This is fulfilled through an electrostatic separation process that separates small particles according to mass in a low-energy charged field. In

particular, a process of thermal and chemical separation (pyrolysis) enables the recovery of ultrapure silicon and other lost metals during the recycling of PV cells at their EoL.

In the next section, the authors illustrate the proposed design methods for PV panel recycling, discussing the optimal materials, the best geometries and geometric proportions as well as the most convenient geometric tolerances in the couplings between the layers and the components that comprise the panel, to achieve longer durability of the PV panels and to attain the most current, efficient and effective solutions for recycling at their EoL.

### 3. Effective PV Panel Design Methods for Recycling at EoL

The analysis of the data from the scientific literature and that which is collected in the *Laboratory of Renewable Energy, Embedded System and Information Processing* of Mohammed First University, Oujda, Morocco, allowed the authors to establish a list of the most common causes of damage/failure of PV panels, in the order of frequency with which they occur, and a list of critical elements for proper PV panel maintenance.

Starting from these data, referring to a standard PV panel (Figure 8), the authors propose a repeatable effective design method (for durability and for recycling) with which it is possible to identify the optimal materials, the best geometry and the most efficient geometric and dimensional tolerances in the couplings between layers as well as between layers and the frame in order to extend panel durability without decreasing efficiency.

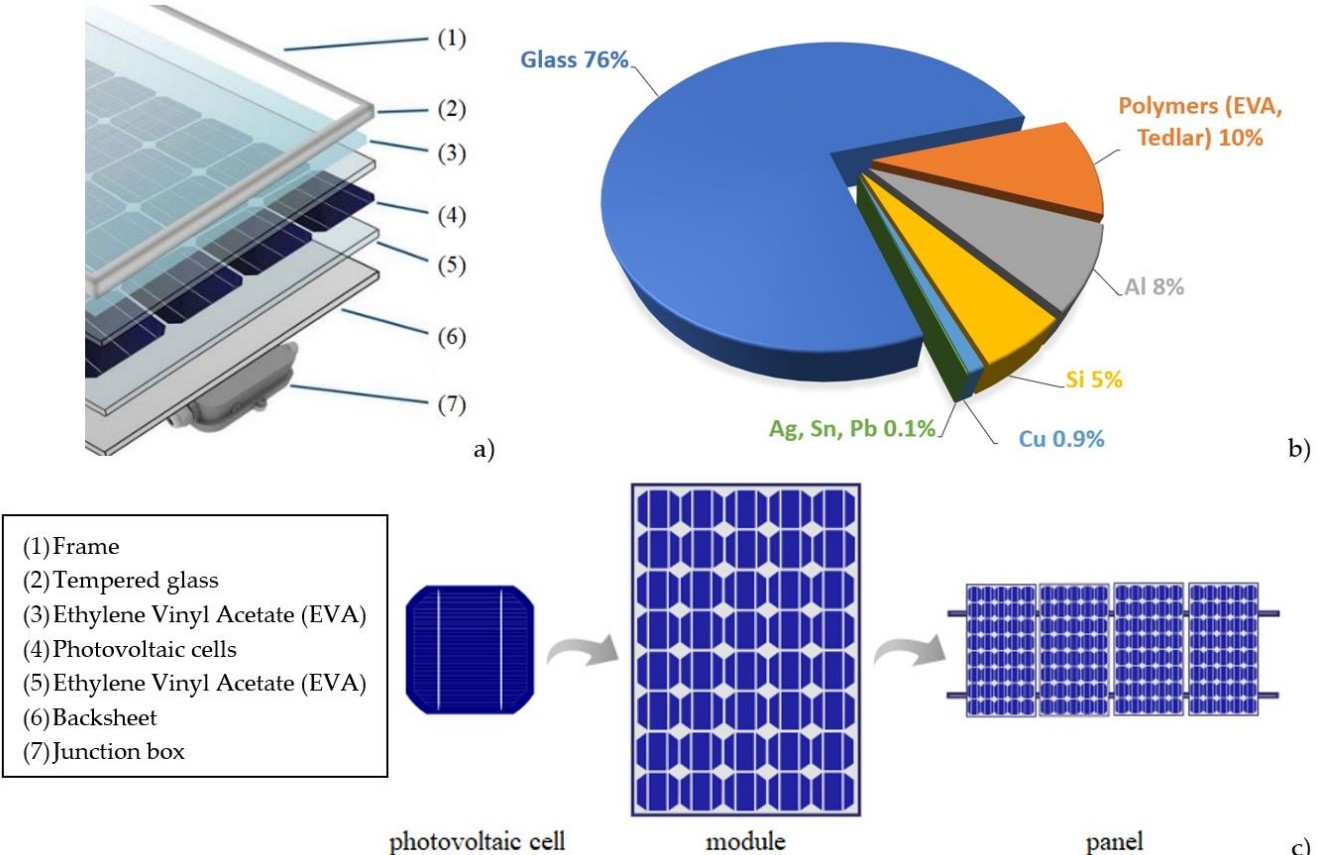

**Figure 8.** (**a**) Standard PV panel structure (source [20]); (**b**) Standard PV panel composition; (**c**) PV panel structure.

In particular, a design for the durability of standard PV panels was carried out by impact, thermal solicitations and fatigue numerical Finite Element (FE) analyses in relation to the most common causes of PV panel damage/failure that were identified (Table 2).

**Table 2.** Recyclable materials for PV panel components and their physical and mechanical characteristics.

| Layer | Material (Completely Recyclable) | Thickness h [mm] | Young's Modulus E [GPa] | Poisson Coefficient v [-] | Thermal Conductivity [W/m °C] | Specific Heat Capacity [J/kg °C] | Density [kg/m³] |
|---|---|---|---|---|---|---|---|
| Frame (1) | Aluminum | 20 ÷ 40 | 69 | 0.31 ÷ 0.34 | 204 | 996 | 2707 |
| Cover (2) | Tempered Glass | 3 ÷ 7 | 70 | 0.22 | 1.8 | 500 | 3000 |
| EVA (3) and (5) | Plastic Material | 0.45 ± 0.05 | 0.015 ÷ 0.08 | 0.48 ÷ 0.49 | 0.35 | 2090 | 960 |
| Photovoltaic Cells (4) | Copper Silicon Silver | 0.4 ± 0.1 | 115 131 83 | 0.33 ÷ 0.36 0.26 ÷ 0.28 0.37 | 148 | 677 | 2330 |
| Backsheet (6) | Tedlar | 0.1 ± 0.05 | 2.1 ÷ 2.6 | | 0.2 | 1250 | 1200 |

### 3.1. Design for Recycling

The International Energy Agency (IEA) emphasizes that PV panels must be designed to return the embedded raw materials or, at least, to provide secondary raw materials that can be entirely used for other applications.

The requirements assessed as critical by the IEA Photovoltaic Power Systems Programme (PVPS) are functionality, longevity, durability, reliability and cost. The Design for Recycling (DfR) must support and improve these aspects.

The first action to be implemented is to eliminate, or at least minimize, the project materials that are difficult to recycle and that are non-reversible adhesives. The composition of the backsheet deserves particular attention, which represents the last layer at the bottom of the photovoltaic solar panel, consisting of a polymer or a combination of polymers. For this, a provision should be made for the use of totally recyclable polymers.

The use of encapsulants should be minimized to facilitate the disassembly of the modules. The use of appropriate sealants in the aluminum frame will allow for the separation of the EoL modules without damaging the components.

In the second column, Table 2 lists the materials for each PV panel component, identified by the authors, which today are completely recyclable. Table 2 also reports the materials' characteristics of density, Young's modulus, Poisson coefficient, stiffness, thermal conductivity and the specific heat capacity used in the parametric FE panel simulation in the next section.

Two main types of PV panels (apart from the PV panels with amorphous silicon that are easily recyclable and that are not treated), which require different recycling approaches, can be traced. Both types, silicon-based panels and thin-film panels, can be recycled using separate industrial processes. Currently, silicon-based panels are the most common [21]. Silicon PV panels are composed mainly of glass, plastic and aluminum, and these are three materials that are recycled in large quantities.

The recycling process of silicon-based PV panels initially involves the disassembly of the actual product to obtain separation between the aluminum and glass parts. Almost all glass can be reused (99.5%), and all external metal parts are employed to reshape cell frames. The remaining materials must be treated at 500 °C in a special heat treatment unit in order to make it easier for cellular elements to bond. Due to the heat, the plastic evaporates, leaving the silicon cells ready to be further processed. The supporting technology ensures that the plastic is not wasted and that it can be reused as a heat source for further thermal processing. After the heat treatment, the hardware is physically separated. As a result, 95% of this can be easily reused, and the rest is further refined. Silicon particles, referred to as "wafers", are eliminated with acid. Wafers that become broken are melted to be reused to produce new silicon modules, with a recycling rate of 98% of the silicon material.

In Figure 9, the authors summarize a logical scheme of the recycling process of PV modules, considering the mechanical, thermal and chemical processes with which it is

possible as well as current technologies used to completely recover all the components at their EoL [22,23]. Two main phases of the diagram are the thermal and chemical processes:

- The **thermal process** must occur at relatively low temperatures (≤60 °C) because, in addition to bringing a minimum energy expenditure and therefore a low environmental impact, this process allows for the avoidance of the thermal degradation of the plastic, allowing, in this way, the recovery of the polymeric materials contained in the panel. In addition, by containing emissions into the atmosphere, it makes it easier to obtain authorization to build any treatment plants.
- The **chemical process**, on the other hand, is the most important step in the recycling process because the chemical treatment conditions must be precisely adjusted to achieve the required level of purity of the recovered silicon. By following quality control, pure silicon in the form of powders can be used for the production of new photovoltaic cells and consequently of new modules.

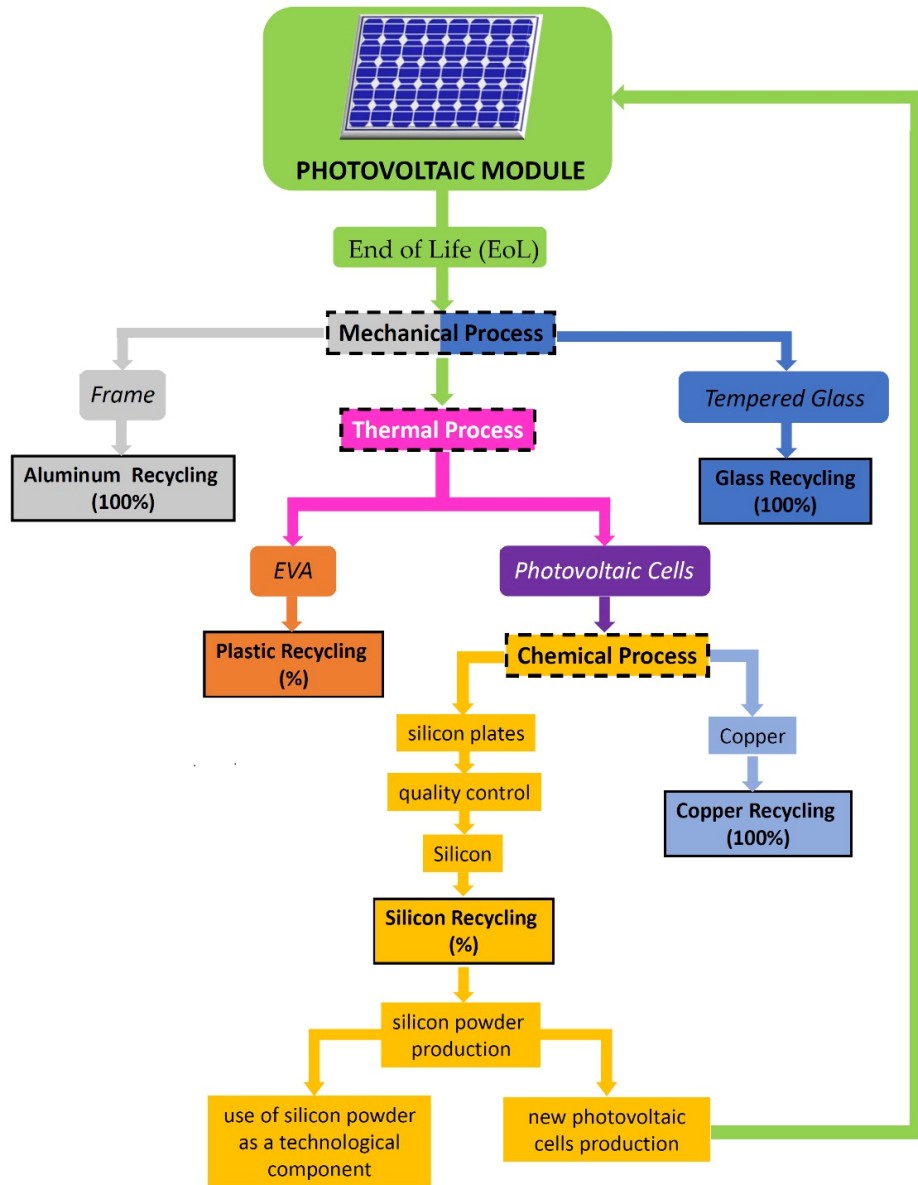

**Figure 9.** Logical scheme for silicon-based PV module total recycling.

The recycling process starts with mechanical disassembly, with which the frame, the tempered glass and the PV module are separated. Then, a thermal process isolates the photovoltaic cells by exploiting the softening of the polymer layers in EVA. Subsequently,

through a chemical process, the materials located inside each cell (silicon and copper) are recovered.

Silicon is a critical raw material that is difficult to completely recover because it is printed in copper circuits in the PV cell and not in a pure state, and it is 'doped' by chemical additives. Therefore, in order to be able to free it, it is necessary to resort to hydrometallurgy processes (to date, they are still particularly expensive). Hydrometallurgy is a process of extracting metals from minerals by dissolving minerals in an aqueous phase and subsequently recovering the metal. Compared to other commonly used techniques, this process has the advantage of being selective, being economical and having a low environmental impact. However, for the complete recyclability of the PV module, its recovery is fundamental.

Adopting the logical scheme above for a very common standard PV module of 21 kg, we can obtain: 15 kg of glass; 2.8 kg of plastic material; 2 kg of aluminum; 1 kg of silicon powder; and 0.14 kg of copper, as shown in Figure 10.

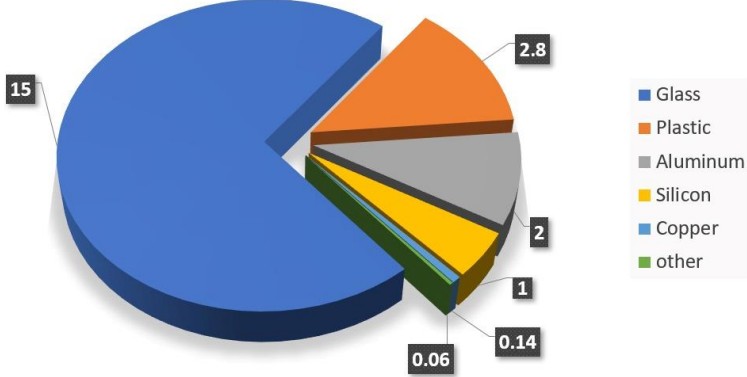

**Figure 10.** Recycling material from a 21 kg commercial PV panel.

The costs and benefits of recycling, especially when externality costs resulting from environmental pollution are considered, are of difficult estimation. By the quantification of the private and externality costs and benefits of recycling crystalline silicon (c-Si) PV panels, it is possible to evaluate the costs of recycling, as performed by Markert et al. [24]. Table 3 reports the costs of recycling, the material value and the net profit for a standard PV module of 21 kg. In particular, the total cost of recycling [€] was obtained from the product of the weight material and the costs of recycling, and the total material value was the product of the weight and material value taken from the work of Markert et al. The net profit was the difference between the total material value and the total cost of recycling.

**Table 3.** Cost of recycling, material value and net profit.

| Material | Weight [kg] | Costs of Recycling [€/kg] | Total Cost of Recycling [€] | Material Value [€/kg] | Total Material Value [€] | Net Profit [€] |
|---|---|---|---|---|---|---|
| Tempered glass | 15 | 2.54 | 38.1 | 150 ÷ 525 | 2250 ÷ 7875 | 2211.9 ÷ 7836.9 |
| Plastic | 2.8 | 0.22 | 0.616 | 3.08 ÷ 5.88 | 8.62 ÷ 16.46 | 8 ÷ 15.84 |
| Aluminum | 2 | 6.62 | 13.24 | 0.94 | 1.88 | −11.36 |
| Silicon | 1 | 3.04 | 3.04 | 25 ÷ 30 | 25 ÷ 30 | 21.96 ÷ 26.96 |
| Copper | 0.14 | 32.17 | 4.5 | 0.14 ÷ 1.12 | 0.02 ÷ 0.16 | −4.48 ÷ −4.34 |

### 3.2. Design for Durability

The standards IEC 61215-1/-1-1/2:2016, ISO 61730-1/2:2016, ISO 9001, ISO 14001, ISO 50001 (Table 4) establish that PV panels and PV modules must be guaranteed for 25 years. They set the tests that they must pass and the minimum values of resistance and performance that they must have (Tables 4–6). In particular, they set the limits of

electrical (Table 5) and mechanical (Table 6) resistance for crystalline cells. The electrical and mechanical properties (Tables 5 and 6) are evaluated under standard test conditions, with irradiation of 1000 W/m$^2$, module temperatures of 25 °C and AM 1.5.

**Table 4.** Certifications and warranty.

| Type of Certifications | Reference Standards |
|---|---|
| General certifications | IEC 61215-1/-1-1/2:2016, ISO 61730-1/2:2016 ISO 9001, ISO 14001, ISO 50001 |
| Ammonia test | IEC 62716 |
| Test corrosion salt spray | IEC 61701 |
| Resistance to fire modulus | Class C, Fire Class 1 (Italy) |
| Product warranty | 25 years |
| Guarantee on the yield of P$_{max}$ (metrological tolerance ±3%) | 25 warranty years |

**Table 5.** Electrical properties (STC) of PV panels.

| Properties | | Value |
|---|---|---|
| Maximum power Pmax | [W] | 355 |
| Tension MPP Vmpp | [V] | 35.7 |
| Current MPP Impp | [A] | 9.95 |
| No-load voltage Voc | [V] | 41.4 |
| Short circuit current Isc | [A] | 10.65 |
| Module performance | [%] | 20.7 |
| Operating temperature | [°C] | $-40 \div +90$ |
| Maximum system voltage | [V] | 1.000 |
| Maximum reverse current | [A] | 20 |
| Power tolerance | [%] | $0 \div +3$ |

**Table 6.** Mechanical properties of PV panels.

| Properties | Value |
|---|---|
| Cell measurement | 161.7 × 161 × 7 mm |
| Cell collector bars | 12 (multiwire collector bar) |
| Dimensions (L × P × H) | 1686 × 1016 × 40 mm |
| Mechanical load | 6000 Pa (pressure) 5400 Pa (wind) |
| Weight | 17.1 kg |
| Connector type | MC4/MC |
| Junction scale | IP68 with 3 diodi by bypass |
| Connection cable | 2 × 1000 mm |
| Front cover | High transmittance tempered glass |
| Frame | Anodized aluminum |

Excluding the failures that occur immediately after construction (infant failure) and the midlife failure, which can be shown to affect the failure of the PV panels with negligible percentages, the main causes that lead more or less slowly and/or instantly to the drastic decrease in the efficiency of the panel and/or its decommissioning were collected and listed by the authors in Table 7. Figure 11 show three main causes. The authors divided the causes into three order of frequency (Low, frequency <1%; Medium, frequency >1%; and High, frequency >5%). For each of these damage/failure causes identified, the more effective ways in which it is possible to avoid this type of damage are reported in the last column [25].

**Table 7.** Damage/failure causes in PV panels.

| Failure Cause | Estimated Frequency | Description | Method to Avoid Damage |
|---|---|---|---|
| *Falling debris, breakage for impacts and micro-cracks* | High | Scratches and breakage due to falling hail, debris, including whole branches, acorns, twigs, etc. Consequences of incorrect production, shipment and installation. | Produce small PV panels/small PV modules. Make bars in the panels. |
| *Internal corrosion (rust) and delamination* | High | Rust occurs when moisture penetrates the panel. Moisture leads to corrosion that becomes visible as a result of darker stains on the panel. | Make airtight or watertight panels by vacuum-rolling the components of the panels (the glass layer, solar cells and EVA sheets). |
| *Water damage* | Medium | Water damage caused by deterioration or old age. | Panel completely sealed. |
| *Hotspot* | Medium | Spots on PV panels caused mainly by poorly welded connections or as a result of a structural defect in the cells. | Control of the absolute quality of the cells during assembly. |
| *Contamination of snail traces* | Low | It is a defect related to the discoloration of the panel. The causes that generate this defect are multiple, including the formation of microscopic cracks in the panel and the use of silver paste of defective frontal metallization. | Limit the mechanical and thermal stress of the panel (even during installation). Silver paste quality controller. |
| *PID (Potential Induced Degradation) effect* | Low | Due to a voltage difference between the panel (grounded) and the grounding. | Monitor the voltage difference. |

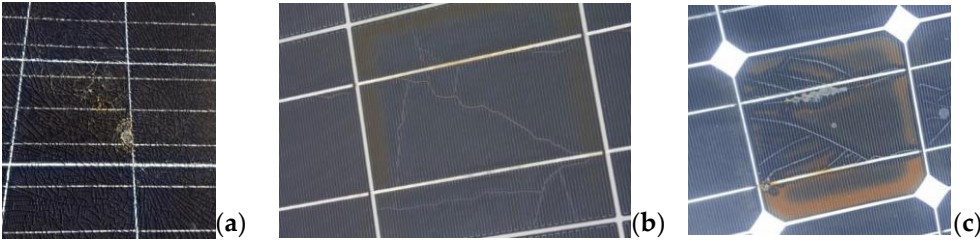

(a)  (b)  (c)

**Figure 11.** (**a**) Falling debris and breakage for impacts; (**b**) water damage; (**c**) contamination of snail traces.

All elements, except for the junction box and the frame, are inserted at the time of assembly into a laminator, whose temperature reaches 145°. In a vacuum process, EVA is heated and fixes the parts together, isolating the PV cells to preserve them from external deterioration agents. In this process, it is very important that the last layer of EVA is perfectly adherent and positioned correctly, since, if of low quality and/or positioned incorrectly, it can cause the formation of small air bubbles between the layers that can affect the correct production of the panel.

Over time, the thermo-mechanical stresses end up leading to the delamination of layers of EVA from the photovoltaic cell layer and the backsheet layer due to the creep phenomenon. In the following sections, an effective numerical method which makes it possible to evaluate this phenomenon and to predict the durability of the panel is described.

### 3.3. Durability Prediction with FE Models

Once the materials are chosen and characterized, multiphysics numerical simulation can provide valuable help to simulate the creep and thermo-mechanical fatigue (TMF) of PV panel components (metal and plastic) with high accuracy. Even with the remarkable variability related to the multiple different operating conditions of the PV panels, the authors wanted to provide a numerical simulation methodology that uses a parametric model appropriately calibrated on the specific operating conditions that allow it to carry out the durability prediction and to establish the best geometries and the most favorable geometric proportions of the PV panel. The results that were found are encouraging and confirm the possibility of reducing time-consuming and costly physical testing in durability predictions.

In the next section, the authors report a case study that confirms the goodness of the proposed method, illustrating in detail the tools and methods that were used.

## 4. PV Panel Thermo-Mechanical Fatigue Analysis Using a Parametric FE Model

In order to assess the most frequent damage/failures reported in Table 6 due to falling debris and breakage from impacts, micro-cracks and the contamination of snail traces and to increase the expected duration of the PV panel, a parametric FE numerical model was developed (Figure 12) and employed in TMF optimization analysis.

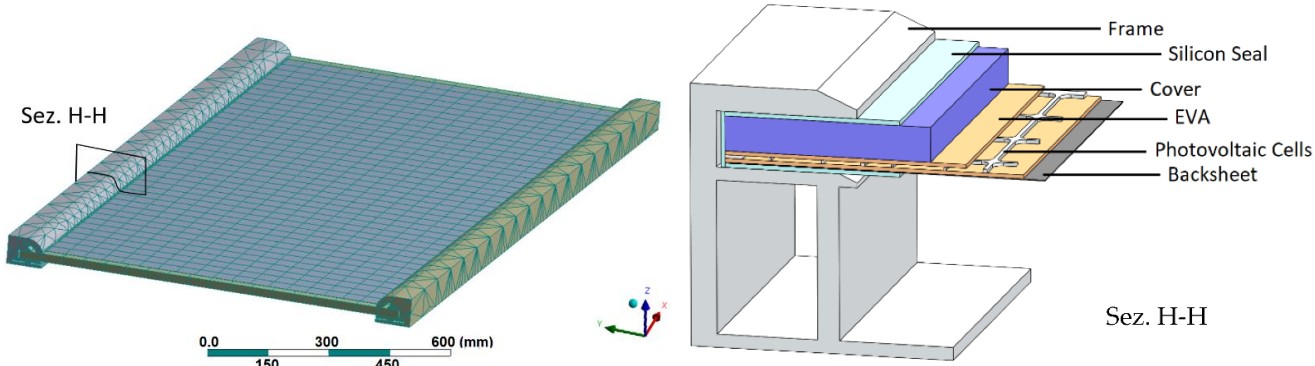

**Figure 12.** Parametric FE model of silicon-based PV panels with support.

The h thicknesses of the different panel layers (Figure 13) and the main overall dimensions (A, B, C, E, F, G, I, L and N) visible in Figure 14 were parameterized within the range of variation, as shown in Table 8. As a consequence of the preliminary study conducted by the authors, a layer of silicone rubber of $h_s$ thickness was predicted in the interface between the layers and the support. This thickness, as we will see, plays an important role in maintaining the adhesion between the layers, limiting the thermo-mechanical stresses and prolonging the life of the PV panel.

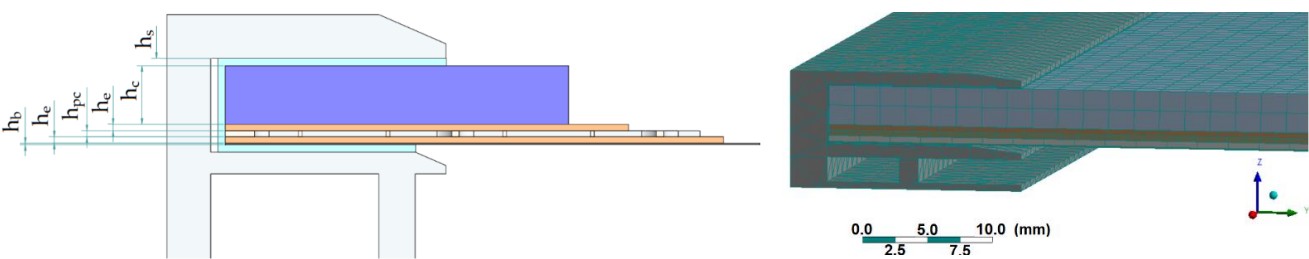

**Figure 13.** Layer thickness parameters in the FE model.

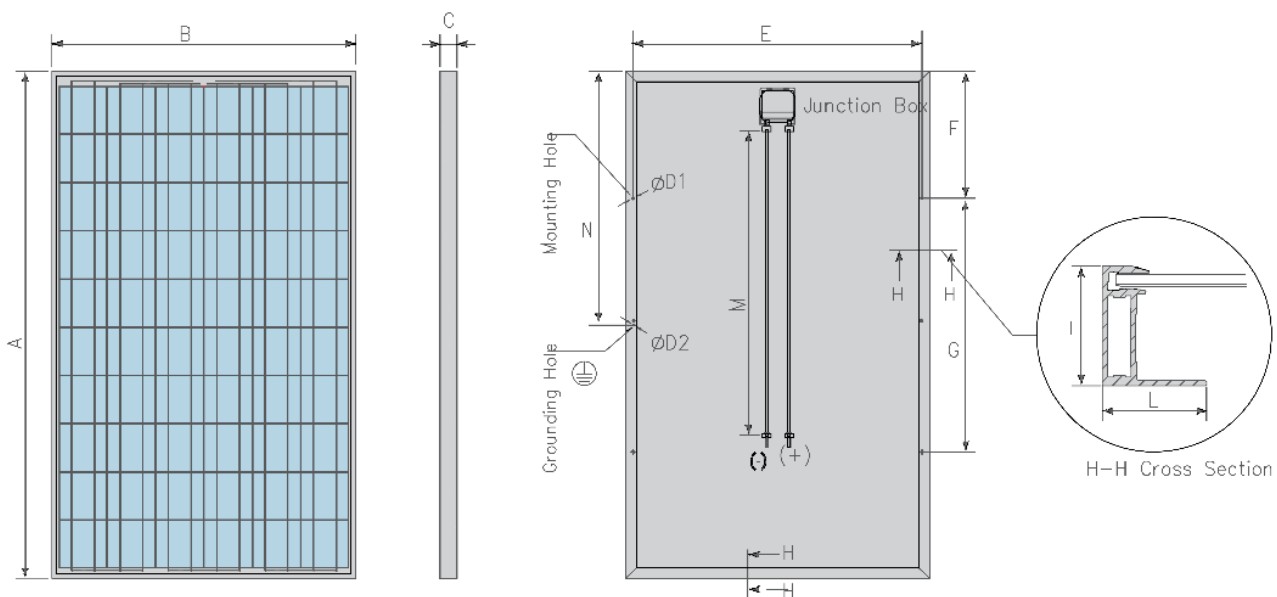

**Figure 14.** Overall dimensional parameters in FE model.

**Table 8.** PV panel dimensional parameters.

| Parameter | Value [mm] | Opt. Value | Parameter | Values [mm] | Opt. Value | Parameter | Values [mm] | Opt. Value |
|---|---|---|---|---|---|---|---|---|
| A | $150 \div 2800$ | **700** | F | $330 \div 430$ | **350** | $h_b$ | $0.1 \div 0.2$ | **0.1 $\pm$ 0.05** |
| B | $150 \div 1700$ | **350** | G | $800 \div 1180$ | **800** | $h_e$ | $0.4 \div 0.5$ | **0.45 $\pm$ 0.05** |
| C | $40 \div 50$ | **40** | I | 40 | **40** | $h_{pc}$ | $0.45 \pm 0.1$ | **0.45 $\pm$ 0.1** |
| $D_1$ | 10 | **10** | L | 35 | **35** | $h_c$ | $3 \div 7$ | **3 $\pm$ 0.05** |
| $D_2$ | 4 | **4** | M | $800 \div 1200$ | **800** | $h_s$ | $0.2 \div 0.4$ | **0.35 $\pm$ 0.1** |
| E | $946 \div 950$ | **946** | N | $830 \div 834$ | **830** | | | |

A DOE optimization on the geometric parameters was carried out by simulating TMF stress with the commercial code ANSYS Thermo-Mechanical Fatigue® vers 18.0.

Tetrahedral ten-node elements were used to calculate the structural deformation of the layer panel with a 6000 Pa pressure (solicitation required by the standards set out in Section 3.2) acting on the top surface of the panel, and a fixed joint was used to lock the panel support on the ground (Figure 12). Static structural simulations and TMF simulations with structural and thermal cyclic loads were repeated in turn, considering all possible combinations between geometric parameters in a DOE analysis [26,27]. The physical and mechanical parameters used in the simulations were those of Table 2. Values of silicone rubber were added to them, as shown in Table 9. The curves S-N of the materials were taken from the literature [28–30].

**Table 9.** Silicone rubber physical and mechanical characteristics.

| Layer | Material (Completely Recyclable) | Thickness h [mm] | Young's Modulus E [GPa] | Poisson Coefficient v [-] | Thermal Conductivity [W/m °C] | Specific Heat Capacity [J/kg °C] | Density [kg/m³] |
|---|---|---|---|---|---|---|---|
| Silicone rubber | Silicone | $0.2 \pm 0.4$ | $0.002 \div 0.007$ | 0.47 | 0.22 | 1350 | 1150 |

In the fatigue calculation, it was assumed that all the PV panel component tensions remained in the elastic field and were low enough. Two cyclic stresses were applied simultaneously to the PV panel:

- a low-frequency mechanical load with a constant amplitude between 0 and 6000 Pa pressure (maximum mechanical load required by the standards set out in Section 3.2) (Figure 15a);
- thermal stress with a daily cyclic frequency and variable amplitude between –40 °C and 90 °C (operating temperature required by the standards set out in Section 3.2) (Figure 15b).

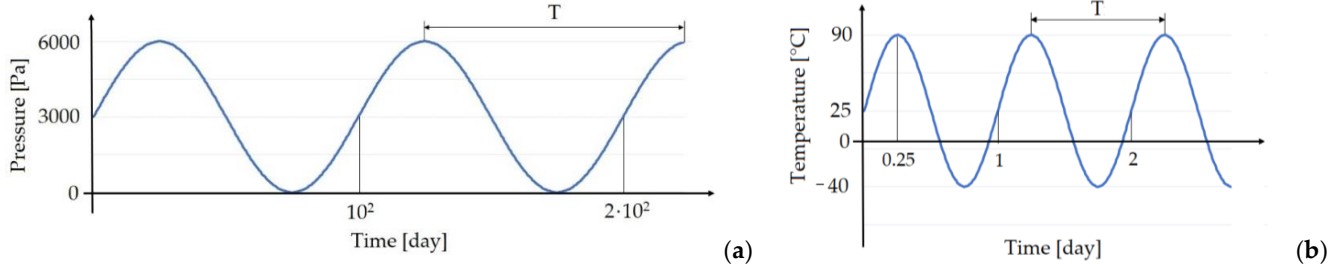

**Figure 15.** Mechanical (**a**) and thermal (**b**) cyclic stress applied on panels in TMF simulation.

The dimension of $0.4 \pm 0.1$ for the thickness of standard PV cells (mono or polycrystalline commercial PV cells) was taken as fixed data, and the thicknesses of each of the other layers, in order to guarantee longer durability with a similar layer stress, were evaluated. Factors of safety and total life cycles by equivalent von Mises stress are shown in Figure 16a and in Figure 16b, respectively. In both TMF simulations, the PV cell layer was found to be the most critical component. However, using a proper thickness ($h_s$) of the silicone rubber layer, it is possible to increase the safety factor by more than 40% (Figure 16c) and the total life cycle (Figure 16d) of the PV panel. The use of a silicone rubber layer also guarantees excellent impact resistance and the finest coupling and insulation conditions. In Table 7, highlighted in bold, the best values of geometric parameters that guarantee the greatest value of the total life are presented.

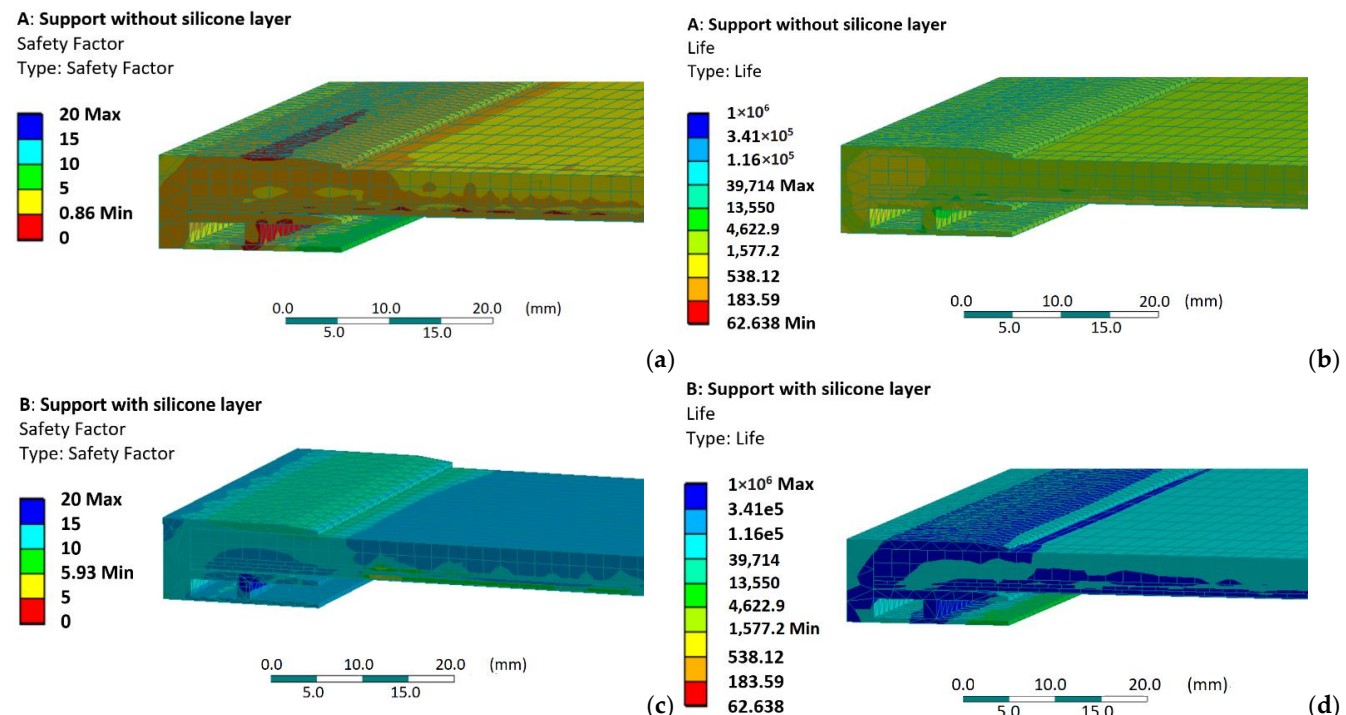

**Figure 16.** (**a**) Safety factor without silicone layer; (**b**) Total life cycle without silicone layer; (**c**) Safety factor with silicone layer; (**d**) Total life cycle with silicone layer.

The dimensioned drawing of the layer support section with dimensional tolerance in the couplings between the layers and the support is shown in Figure 17, and Figure 18 summarizes a qualitative trend of the influence that the thermal variation and the stress range have on the durability of the PV panel (total life index). It is highlighted how thermal stress with a daily stress frequency has a greater influence than mechanical stress of 6000 Pa pressure (maximum mechanical load required by the standards set out in Section 3.2), which was considered in the TMF simulation with a frequency of 10 days.

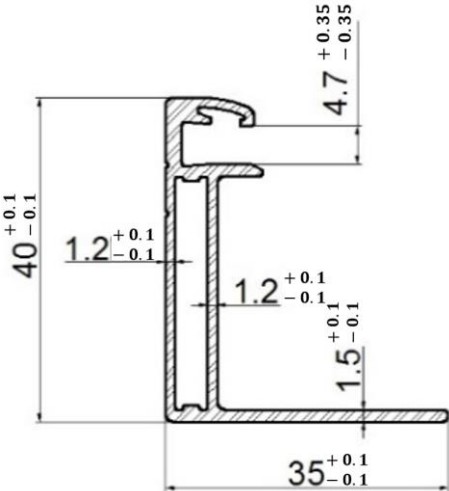

**Figure 17.** Optimized layer support section.

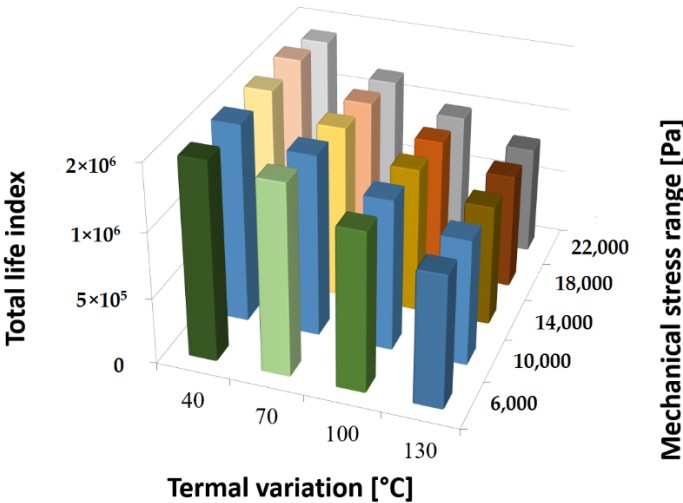

**Figure 18.** Thermal variation and stress range vs. total life index in PV panels.

Other results found by the authors and the relationships with other elements that are critical to the durability of PV panels will be considered by the authors in future work.

To complete the analysis of the most frequent causes of damage/failure, the solicitation of an impulse force with an impact on the edge of the panel (Figure 19) was analyzed. The force modulus was assumed equal to the force to which the panel is subjected when falling from a height of 1.5 m. In this case, it can be seen how, within the assumed panel size range, the greatest stresses occur on the PV cell, and even for the larger dimensions of the panel, it remains below the permissible sigma. The stresses, however, cause micro-cracks that, in the larger panels, are of higher intensity and that affect a greater number of PV cells.

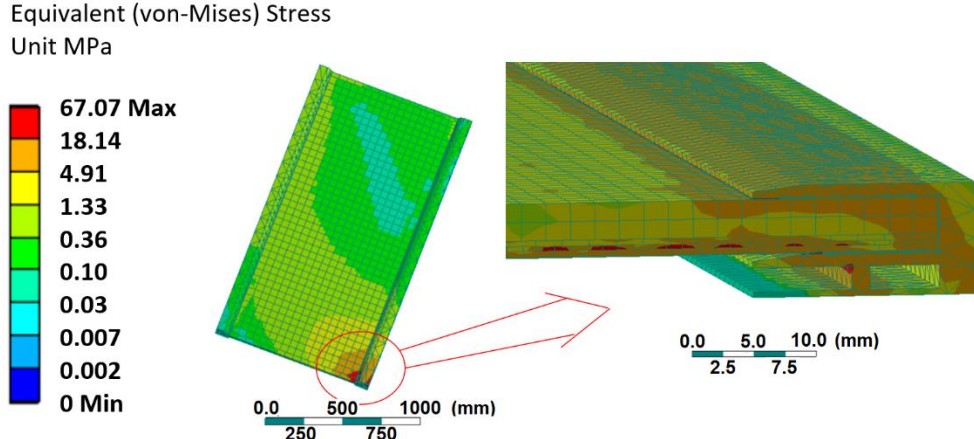

**Figure 19.** Equivalent von Mises stress in the case of impact on an edge of the panel.

Using the proposed design and maintenance methodology, the authors estimate that PV panels can last up to 40 years or more, while still offering only slightly reduced energy production.

## 5. Conclusions

PV technology is one of the most important technologies for the environment, but at the same time, PV panels are considered responsible for an increasingly larger amount of waste per unit of energy compared to various sustainable power generation and electricity technologies. The recycling of PV panels is the main focus of researchers for the mitigation of environmental problems related to EoL treatments and to allow for further increases in opportunities for economic and social development while reducing greenhouse gas emissions through electricity generation.

The analysis carried out in this paper highlights how the materials of PV panels that can be totally recycled (recycling rate 100%) are glass and polymers (encapsulant of the panel -EVA- and backsheet). Polymer encapsulants and backsheets are more difficult to recycle and have a very low commercial value.

Among the remaining materials that comprise the panel, the total recovery of silicon is the most interesting challenge that is still open. To date, only some of the wafer processing techniques allow for the recovery of silicon (high quality) with a recycling rate of 100%. However, the silicon thus recovered is considerably more valuable and improves the recycling economy.

The analysis that was carried out has led to the conclusion that, to date, by making recycling economically advantageous or, at least, encouraging it with government contributions, it is possible for the total recycling of panels to their EoL.

However, the study also provides practical design information for the most current, efficient and effective solutions for recycling at the end-of-life (EoL) together with longer durability. Using a TMF analysis performed on a parametric FE model, the authors have shown how it is possible to find the best geometries and geometric proportions and the most convenient geometric and dimensional tolerances in the couplings between the layers and the components that comprise the PV panel. It is shown here how these tools allow one to achieve the most current, efficient and effective solutions for recycling at EoL and to achieve longer durability. The remarkable sensitivity of the durability of PV panels to thermal stress is also evaluated. It was evaluated how the installation of smaller PV panels with proper thicknesses is particularly convenient. Although the initial installation is more expensive, the replacements are simpler and cheaper, and the loss of energy production is reduced in the case of damage to the panel.

**Author Contributions:** Conceptualization, M.C. and A.A.; methodology, M.C., A.A. and B.H.; software, M.C.; validation, M.C., A.A., G.N. and B.H.; formal analysis, M.C. and A.A.; investigation,

M.C. and A.A.; resources, M.C., A.A. and G.N.; data curation, M.C., A.A. and B.H.; writing—original draft preparation, M.C., A.A. and G.N.; writing—review and editing, M.C., A.A., G.N. and B.H.; visualization, M.C. and A.A.; supervision, M.C.; project administration, M.C. and A.A.; funding acquisition, M.C. All authors have read and agreed to the published version of the manuscript.

**Funding:** This work was supported by the University of Catania [PIACERI 2020/22–GOSPEL/UPB: 61722102132].

**Institutional Review Board Statement:** Not applicable.

**Informed Consent Statement:** Not applicable.

**Data Availability Statement:** The corresponding authors will be happy to provide data that are not directly available in the article.

**Acknowledgments:** This paper belongs to a research path funded by the University of Catania (Starting Grant 2020–2022 Linea 3—Progetto NASCAR—Prot. 308811).

**Conflicts of Interest:** The authors declare no conflict of interest with respect to the research, authorship and publication of this article.

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
