# Peer review of "The Design Value for Recycling End-of-Life Photovoltaic Panels"

_applsci, doi:10.3390/app12189092_

Round 1

Reviewer 1 Report

You presented solutions, but no possible impact of these solution. Everybody knows that we have to recycle them, but is important to estimate the impact of these recycling processes.  

Author Response

Dear Editor,

the authors gratefully acknowledge the constructive comments on our manuscript. In the revised version, the amendments to the paper are highlighted using yellow colour. The authors have also better arranged the formatting of the manuscript for some inaccuracies in form and grammar. Now the manuscript was deeply modified according to the reviewers’ comments. The title was slightly modified; Prof.Hajji Bekkay Director of the Laboratory of Renewable Energy, Embedded System and Information Processing, National School of Applied Sciences, Mohammed First University, Oujda 60000, Morocco and guest Editor of this Special Issue “Design for Sustainability and EcoDesign in Renewable Energy Production and Transport" was added among the authors of the work. He provided data on the photovoltaic panels used in the manuscript and he contributed to write the revised version of the article. A new section on the design methodology proposed (Section 3 p. 7) and a new section on the analyses carried out (Section 4 p. 12) were inserted in the manuscript.

Reviewer 1

1

Line 116 – Considering the study of Chowdhury et al. [16]

ü  Authors thank the reviewer for the corrections. The reference 16 now was properly called.

2

Line 121-122 – From [11] it can be estimated -------- It can be estimated [11]

ü  The reference 11 now was properly called.

3

Line 147 – Do you think that silicon-based panels are most common because of the materials that are recycled or because now their production costs are lower? What I mean is that I am not sure that the producers are really thinking in terms of recyclability, but profitability, in this moment. Or, what you want to say is that they are most common in recycling? It is not clear what you mean. ?

ü  The phrase could be misleading. Authors have now specified in the text (line 86 -89 pag. 3) that silicon-based panels are more common because they have lower costs than thin-film-based PV panels.

From a recycling perspective: both types of solar panel (silicon-based and thin-film based) have almost the same materials that can be recycled. So, the material that can be recycled varies by the amount of material used in the two panels. Of course, as mentioned, until now PV panels manufacturers have chiefly thinking in terms of profitability and not in terms of recyclability.

4

Line 164 - .. are processed in a more drastic process manner or way.

ü  The sentence now was corrected (line 164-167 Section 2).

5

Line 232 -238– what is the purpose of these information? What is the link with the infos above? Please explain. And anyway, the presentation of solar panels (Lines 185-232) I think should be better if done before the explanation of recycling process. For more coherence.

ü  Now the manuscript was deeply modified. The description of solar panels is before the recycling process illustration. A new section on the design methodology proposed (Section 3 p.  XX) and a new section on the analyses carried out (Section 4 p. 12) were inserted in the paper. The figure indicated by the reviewer has now been included in this section to explain the methodology.

Reviewer 2 Report

The topic is interesting and the presentation of the content is of good quality. However, after reading through the manuscript, I cannot judge whether it is an original research article or a review paper. If it is an original research article, can the authors please include a dedicated methodology section to explain how the study was conducted? If it is a review article, can the authors clearly state what is the focus of the review and the key findings from the literature? The manuscript reads like a magazine article or textbook chapter with good content to educate the general audience but is not suitable to be published as a scientific journal article. Overall, the authors need to clearly identify the novelty of this work to be considered in the Journal of Applied Sciences. 

Author Response

Dear Editor,

the authors gratefully acknowledge the constructive comments on our manuscript. In the revised version, the amendments to the paper are highlighted using yellow colour. The authors have also better arranged the formatting of the manuscript for some inaccuracies in form and grammar. Now the manuscript was deeply modified according to the reviewers’ comments. The title was slightly modified; Prof. Hajji Bekkay Director of the Laboratory of Renewable Energy, Embedded System and Information Processing, National School of Applied Sciences, Mohammed First University, Oujda 60000, Morocco and guest Editor of this Special Issue “Design for Sustainability and EcoDesign in Renewable Energy Production and Transport" was added among the authors of the work. He provided data on the photovoltaic panels used in the manuscript and he contributed to write the revised version of the article. A new section on the design methodology proposed (Section 3 p. 7) and a new section on the analyses carried out (Section 4 p. 12) were inserted in the paper.

Reviewer 2

1

The topic is interesting and the presentation of the content is of good quality. However, after reading through the manuscript, I cannot judge whether it is an original research article or a review paper. If it is an original research article, can the authors please include a dedicated methodology section to explain how the study was conducted? If it is a review article, can the authors clearly state what is the focus of the review and the key findings from the literature? The manuscript reads like a magazine article or textbook chapter with good content to educate the general audience but is not suitable to be published as a scientific journal article. Overall, the authors need to clearly identify the novelty of this work to be considered in the Journal of Applied Sciences.

ü  Authors thank the reviewer and agree with his analysis. In the previous version of the manuscript, authors had not explained well the methodology followed for lack of time. Now the manuscript was deeply modified. A new section on the design methodology proposed (Section 3 p. 7) and a new section on the analyses carried out (Section 4 p. 12) were inserted in the paper. The study has identified the optimal materials, the best geometries and geometric proportions and the most convenient geometric and dimensional tolerances in the couplings between the layers and the components that make up the panel. Finally, a logical scheme was proposed that allows the total recycling of materials at the end of their life for the panel designed and built in the optimized way.

Reviewer 3 Report

The paper deals with recycling challenges of solar panels.

Most of the information stated in the paper is common knowledge.
While most figures have been taken from references, it is not clear whether Figure 5 is the authors own contribution or also taken from another source. Please explain.

The values in the tables seem to be taken from other sources as well.

Hence, I cannot see where the novelty or significant contribution of this paper lies.
Please explain this in detail, highlighting your own contribution where appropriate.

With the exception of the last sentence, the abstract reads like an introduction.
The abstract should mainly summarise your contribution and only briefly mention background information.

Figure 5: What does the cumulative PV capacity mean here? Is it just tons of waste translated to GW of waste PV capacity?

Table 1, row 6: Why are there two percentages (78 and 10)?

Table 3: In English, use commas as thousands separators, not decimal points.

While the English language is syntactically correct, some sentences sound strange. Proofreading by a native English speaker is therefore recommended.

Author Response

Dear Editor,

the authors gratefully acknowledge the constructive comments on our manuscript. In the revised version, the amendments to the paper are highlighted using yellow colour. The authors have also better arranged the formatting of the manuscript for some inaccuracies in form and grammar. Now the manuscript was deeply modified according to the reviewers’ comments. The title was slightly modified; Prof. Hajji Bekkay Director of the Laboratory of Renewable Energy, Embedded System and Information Processing, National School of Applied Sciences, Mohammed First University, Oujda 60000, Morocco and guest Editor of this Special Issue “Design for Sustainability and EcoDesign in Renewable Energy Production and Transport" was added among the authors of the work. He provided data on the photovoltaic panels used in the manuscript and he contributed to write the revised version of the article. A new section on the design methodology proposed (Section 3 p. 7) and a new section on the analyses carried out (Section 4 p. 12) were inserted in the paper.

Reviewer 3

1

The paper deals with recycling challenges of solar panels.

Most of the information stated in the paper is common knowledge. While most figures have been taken from references, it is not clear whether Figure 5 is the authors own contribution or also taken from another source. Please explain.

The values in the tables seem to be taken from other sources as well.

Hence, I cannot see where the novelty or significant contribution of this paper lies. Please explain this in detail, highlighting your own contribution where appropriate.

Figure 5: What does the cumulative PV capacity mean here? Is it just tons of waste translated to GW of waste PV capacity?

ü  Authors thank the reviewer. The values in Figure 5 are estimated values. On the basis of the many data analyzed and on the basis of the evaluation carried out with this simulation, the authors have foreseen the amount of waste produced in the next 30 years according to different scenery of duration of the photovoltaic panels. These estimate tons of waste were translated into GW producible (section XX p. XX). The cumulative PV capacity was removed from the diagram.

The novelty lies in the design method proposed by the authors that allows to evaluate the durability and performance of the panel in relation to the materials, the best geometries and geometric proportions.

2

With the exception of the last sentence, the abstract reads like an introduction. The abstract should mainly summarise your contribution and only briefly mention background information.

ü  A substantial part of the abstract has been rewritten. The last part of the abstract, as suggested by the reviewer, now summarizes the original contribution of the authors on the issue dealt with.

3

Table 1, row 6: Why are there two percentages (78 and 10)?

ü  Now, the data of Table 1 has been reported in the diagram of Figure 10 (Section 3 p. 10).

4

Table 3: In English, use commas as thousands separators, not decimal points.

Separators in Table 3 were corrected.

5

While the English language is syntactically correct, some sentences sound strange. Proofreading by a native English speaker is therefore recommended.

ü  The text of the manuscript was now revised by a native English speaker.

Reviewer 4 Report

This study is dedicated to exploration of solar panels recycling methods. Here are my thoughts and comments to the content:

1.The abstract. "The greatest production of electricity no longer comes from conventional sources....". Research articles should be objective. No need to start with such false argument. The major part of global energy generation is fossil-based and this situation will not change in the coming decade.

2.Introduction. It is not a representative overview. Today Green Deal is a big question due to Russian-Ukraine conflict. There are plans to start using coal for energy generation due to the lack of natural gas from Russia and there is no "renewable" solution for this problem in the coming years. This situation should be discussed, instead of 2019 document.

3.Lines 89-91. Another not objective sentence. There are several reasons for such fast scaling of renewable energy. And one of the most important thing is the implementation ETSs, carbon taxes and various instruments like FIT, FIP etc.

4.Fig 5. Source of data? By the way, it is necessary to add the source of information in the caption of each figure and table. It is not author's material.

5.I don't understand the aim of this study. Authors are trying to show some achievements in solar panels utilization, but this information is not new. There are also no new conclusions or recommendations.

Taking into account that this manuscript is submitted to the APPLIED sciences journal, i recommend to reject it. Presented information is too superficial to be useful for practice.

Author Response

Dear Editor,

the authors gratefully acknowledge the constructive comments on our manuscript. In the revised version, the amendments to the paper are highlighted using yellow colour. The authors have also better arranged the formatting of the manuscript for some inaccuracies in form and grammar. Now the manuscript was deeply modified according to the reviewers’ comments. The title was slightly modified; Prof. Hajji Bekkay Director of the Laboratory of Renewable Energy, Embedded System and Information Processing, National School of Applied Sciences, Mohammed First University, Oujda 60000, Morocco and guest Editor of this Special Issue “Design for Sustainability and EcoDesign in Renewable Energy Production and Transport" was added among the authors of the work. He provided data on the photovoltaic panels used in the manuscript and he contributed to write the revised version of the article. A new section on the design methodology proposed (Section 3 p. 7) and a new section on the analyses carried out (Section 4 p. 12) were inserted in the paper.

Reviewer 4

1

This study is dedicated to exploration of solar panels recycling methods. Here are my thoughts and comments to the content:

1. The abstract. "The greatest production of electricity no longer comes from conventional sources....". Research articles should be objective. No need to start with such false argument. The major part of global energy generation is fossil-based and this situation will not change in the coming decade.

ü  Authors thank the reviewer. A substantial part of the abstract has been rewritten. The last part of the abstract, as suggested by the reviewer, now summarizes the original contribution of the authors on the issue dealt with.

2

2. Introduction. It is not a representative overview. Today Green Deal is a big question due to Russian-Ukraine conflict. There are plans to start using coal for energy generation due to the lack of natural gas from Russia and there is no "renewable" solution for this problem in the coming years. This situation should be discussed, instead of 2019 document.

ü  In the last version of the manuscript, authors have mentioned the current Russian-Ukraine conflict. Surely this conflict has worsened the situation and the application of the targeted actions envisaged in the Dreen Deal. However, authors do not think that the conflict has definitively affected the "renewable" solution proposed in the Dreen Deal and indeed the lack of natural gas from Russia can act as an incentive for a faster transition to "renewable" solution. With this in mind the methodology presented in the article is fully valid.

3

3. Lines 89-91. Another not objective sentence. There are several reasons for such fast scaling of renewable energy. And one of the most important thing is the implementation ETSs, carbon taxes and various instruments like FIT, FIP etc.

ü  In current lines 29-36 authors rewrite the sentence  mentioned the possibility of to apply photovoltaic panels in industrial basins, irrigation tanks and drinking water tanks, by using the so-called floating photovoltaic (FPV).

4

4. Fig 5. Source of data? By the way, it is necessary to add the source of information in the caption of each figure and table. It is not author's material.

ü  The analyzed data were provided by the Laboratory of Renewable Energy, Embedded System and Information Processing, National School of Applied Sciences, Mohammed First University, Oujda 60000, Morocco.

Figure 5 diagram are estimated values. Authors, on the basis of the many data analyzed and of the evaluation carried out with their simulation, have foreseen the amount of waste produced in the next 30 years according to a different scenery of duration of the photovoltaic panels. The sources of all figures and tables not original in the manuscript were cited using proper caption and references.

5

5. I don't understand the aim of this study. Authors are trying to show some achievements in solar panels utilization, but this information is not new. There are also no new conclusions or recommendations.

Taking into account that this manuscript is submitted to the APPLIED sciences journal, i recommend to reject it. Presented information is too superficial to be useful for practice.

The authors worked hard to improve the manuscript. Now the manuscript was deeply modified.

In the previous version of the manuscript, authors had not explained well the design methodology followed for lack of time. A new section on the design methodology proposed (Section 3 p. 7) and a new section on the analyses carried out (Section 4 p. 12) were now inserted in the manuscript.

Authors wanted to provide guidelines in the design of photovoltaic panels that allow to identify the optimal materials, the best geometries and geometric proportions that maintain high the efficiency of the panels throughout all their operation life and allow the total recycling of materials at the end of their life.

Round 2

Reviewer 1 Report

From my point a view, the economic dimension is totally missing. It is important to recycle, but we must see the costs of recycling. Is there a paragraph with the amount of materials obtained from recycling, it would be interesting to see the market value of these materials and some costs. 

Author Response

The authors gratefully acknowledge the constructive comments on our manuscript. In the revised version, the amendments to the paper are highlighted using yellow colour. The authors have also checked the manuscript to delete some inaccuracies in typos.

Reviewer 1

1

From my point a view, the economic dimension is totally missing. It is important to recycle, but we must see the costs of recycling. Is there a paragraph with the amount of materials obtained from recycling, it would be interesting to see the market value of these materials and some costs.

Authors thank the reviewer. Now in section 3.1 the costs of recycling and the profits that can be obtained are shown (pag. 10 lines 303-313).

Reviewer 3 Report

Thank you for improving the paper.
The contribution is now much clearer.

Since the work comprises a lot of steps, I suggest to add a diagram that shows all the steps of this research, such that readers know what to expect.
This would also help to point out the authors' contribution even more.

Please check the language again for typos etc. (e.g. p. 11, l. 136 "Hight frequency").

Author Response

the authors gratefully acknowledge the constructive comments on our manuscript. In the revised version, the amendments to the paper are highlighted using yellow colour. The authors have also checked the manuscript to delete some inaccuracies in typos. 

Reviewer 3

1

Thank you for improving the paper. The contribution is now much clearer.

Since the work comprises a lot of steps, I suggest to add a diagram that shows all the steps of this research, such that readers know what to expect. This would also help to point out the authors' contribution even more.

Please check the language again for typos etc. (e.g. p. 11, l. 136 "Hight frequency").

Authors thank the reviewer. Now in the Introduction section (pag. 4 lines 117-120) a diagram that shows all the steps of this research is added for more clearness.

Authors checked the manuscript to delete any residues typos.

Reviewer 4 Report

I have to agree that now it is absolutely new article, which seems suitable for publication.

I have only one minor comment. There are many figures from other sources. It is crucial to specify the source of information in the caption of figures. Otherwise, it will look like a plagiarism. Moreover, it is necessary to check the possibility of using these figures in a such manner.

Author Response

The authors gratefully acknowledge the constructive comments on our manuscript. In the revised version, the amendments to the paper are highlighted using yellow colour. The authors have also checked the manuscript to delete some inaccuracies in typos. 

Reviewer 4

1

I have to agree that now it is absolutely new article, which seems suitable for publication.

I have only one minor comment. There are many figures from other sources. It is crucial to specify the source of information in the caption of figures. Otherwise, it will look like a plagiarism. Moreover, it is necessary to check the possibility of using these figures in a such manner.

Authors thank the reviewer. Now in the manuscript there are only 2 figures (Figure 4 and Figure 8a) taken from other authors. In both cases the source of information was specified in the caption of figures as well as in the text of the manuscript.
